# The impact of lockdown enforcement during the SARSCoV-2 pandemic on the timing of presentation and early outcomes of patients with ST-elevation myocardial infarction

Ofer Kobo[1]☉*, Roi Efraim[2]☉, Majdi Saada[1], Natalia Kofman[3,4], Ala Abu Dogosh[5], Yigal Abramowitz[5], Doron Aronson[2,6], Sa'ar Minha[3,4], Ariel Roguin[1,6], Simcha R. Meisel[1,6]

1 Department of Cardiology, Hillel Yaffe Medical Center, Hadera, Israel, 2 Department of Cardiology, Rambam Health Care Campus, Haifa, Israel, 3 Department of Cardiology, Shamir Medical Center, Zerifin, Israel, 4 Sackler Faculty of Medicine, Tel-Aviv University, Tel-Aviv, Israel, 5 Department of Cardiology, Soroka University Medical Center, Faculty of Health Sciences, Ben Gurion University of the Negev, Beer-Sheva, Israel, 6 Rappaport - Faculty of Medicine, Technion - Israel Institute of Technology, Haifa, Israel

☉ These authors contributed equally to this work.
* Ofermkobo@gmail.com

**Data Availability Statement:** The data underlying the results presented in the study are available

## Abstract

### Introduction

Early reports described decreased admissions for acute cardiovascular events during the SarsCoV-2 pandemic. We aimed to explore whether the lockdown enforced during the SARSCoV-2 pandemic in Israel impacted the characteristics of presentation, reperfusion times, and early outcomes of ST-elevation myocardial infarction (STEMI) patients.

### Methods

A multicenter prospective cohort comprising all STEMI patients treated by primary percutaneous coronary intervention admitted to four high-volume cardiac centers in Israel during lockdown (20/3/2020–30/4/2020). STEMI patients treated during the same period in 2019 served as controls.

### Results

The study comprised 243 patients, 107 during the lockdown period of 2020 and 136 during the same period in 2019, with no difference in demographics and clinical characteristics. Patients admitted in 2020 had higher admission and peak troponin levels, had a 2.4 fold greater likelihood of Door-to-balloon times> 90 min (95%CI: 1.2–4.9, p = 0.01) and 3.3 fold greater likelihood of pain-to-balloon times> 12 hours (OR 3.3, 95%CI: 1.3–8.1, p<0.01). They experienced higher rates hemodynamic instability (25.2% vs 14.7%, p = 0.04), longer hospital stay (median, IQR [4, 3–6 Vs 5, 4–6, p = 0.03]), and fewer early (<72 hours) discharge (12.4% Vs 32.4%, p<0.001).

upon request because they contain potentially identifying details about the patients. As the number of STEMI admissions in each institute is limited, the date and time of admission, gender and age are enough to identify patients. The Hillel Yaffe Helsinki Committee imposed these data sharing restrictions. Data requests should be sent to the Ms Naama Amsalem, Cardiology Research Coordinator, Hillel Yaffe Hospital-NaamaA@hy. health.gov.il.

**Funding:** The author(s) received no specific funding for this work.

**Competing interests:** The authors have declared that no competing interests exist.

## Conclusions

The lockdown imposed during the SARSCoV-2 pandemic was associated with a significant lag in the time to reperfusion of STEMI patients. Measures to improves this metric should be implemented during future lockdowns.

## Introduction

Although the pathophysiological mechanism is not fully known, the incidence of acute myocardial infarctions is expected to increase during times of stress [1–4]. In addition to the economic crisis precipitated by the SARS-CoV-2 pandemic, it impacted on the wellbeing of the global population by imposing social restrictions associated with anxiety and stress. However, one of the lockdown consequences was the withdrawal of patients and a tendency to avoid referral to hospitals. Early reports suggested fewer Acute Coronary Syndrome (ACS)-related admissions [5–9] with longer times from symptom onset to medical contact [7, 10]. On March 20th, due to the rising SARS-CoV-2 morbidity (S1 Table), a lockdown was enforced in Israel. We aimed to examine the impact of the government-imposed lockdown in Israel on the characteristics of presentation, reperfusion times and early outcomes of ST-elevation myocardial infarction (STEMI) patients.

## Methods

The current analysis is based on a multicenter prospective cohort from four high-volume cardiac centers in Israel- Hillel Yaffe Medical Center, Rambam Healthcare Campus, Shamir Medical Center and Soroka Medical Center. All participating centers are university-affiliated hospitals with coronary catheterization-laboratories that provide a 24/7 on-call primary PCI (PPCI) service. The study was conducted according to the Declaration of Helsinki, with informed consent being waived due to the observational nature of the study.

Data of all consecutive patients presenting with STEMI, during the lockdown period in Israel, March 20th–end of April 2020 were entered into a dedicated database and compared with retrospective data from the same period in 2019. The Inclusion criterion was a diagnosis of STEMI on admission. The main exclusion criterion was age<18.

The primary endpoints were door-to-balloon(D2B) and pain-to-balloon time intervals (P2B). Secondary endpoints included in-hospital mortality, failure to achieve post-PCI TIMI3 flow, moderately- or severely-reduced left ventricular ejection fraction(<40%) at discharge, need for mechanical ventilation, hemodynamic instability during hospitalization, length of hospital stay, and baseline and peak high sensitivity-troponin level.

Reperfusion times, as well as post PCI TIMI flow were drawn from the PCI report. Hemodynamic instability was defined as a need for vasopressors, cardiogenic shock, or use of mechanical circulatory support.

Multivariable logistic regression models were used to examine the association between 2020 admission and delayed reperfusion intervals. Models were adjusted to baseline demographics and risk factors with 2019 admissions serving as a reference group.

Categorical variables were presented as frequencies and percentages and compared using Pearson's chi-square test or the Fisher-exact test. Continuous variables were presented as median and interquartile range or mean and standard deviation compared using the T-test or the Mann-Whitney U test, as appropriate. A p-value<0.05 was considered statistically

significant. Data analysis was performed using IBM SPSS Statistics for Windows (Version 25.0, Armonk, NY).

## Results

Two-hundred and forty- three patients were included in the study including 136 during March 20-April 30 2019 and 107 during March20 -April 30 2020 (22% decrease). Overall, 7 patients did not undergo PCI (three in 2019, four in 2020, p = 0.37). Only one SARS-CoV-2 positive patient was included in the study. Data on reperfusion time was available for all patients. There were no significant differences in patient demographics and clinical characteristics between the groups. P2B time was significantly longer in 2020, with a higher rate of patients presenting with very-long (>12 hours) P2B intervals and a higher rate of patients failing to meet the D2B<90 minutes guideline constraint. More patients in 2020 developed hemodynamic instability, and their admission and peak troponin levels were higher (p = 0.03, 0.01, respectively). More patients were discharged early (<72 hours) in 2019 as reflected in a significant difference in hospital length of stay (Table 1). No significant difference in other in-hospital outcomes was noted.

In the 2020 group, the adjusted odds for delayed reperfusion times were significantly higher both for D2B>90 min (OR-2.4, 95%CI:1.2–4.9, p = 0.01) and P2B>12 hours (OR3.3, 95% CI:1.3–8.1, p<0.01, Table 2, Fig 1).

## Discussion

The present multicenter study aimed to evaluate the influence of SARS-CoV-2 pandemic and its attendant government restrictions on the timeliness of STEMI presentation and early outcomes. In contrary to the anticipated rise in cardiovascular events [1–4], early reports revealed a decrease in ACS admissions during the SARS-CoV-2 pandemic [5–9]. Correspondingly, we found a significant 22% decrease in the incidence of STEMI presentations during the lockdown period, compared with the same period in 2019. Possible explanations for this decrease were previously suggested [7] and include a change in lifestyle during the lockdown, fear from contacting SARS-CoV-2 patients in the hospitals, dismissing ACS symptoms as viral related, and avoiding any unnecessary burden on the strained medical staff.

Reperfusion times directly affect the clinical outcomes of STEMI patients [11–13]. We observed a 2.4-fold greater likelihood of prolonged D2B (>90 min) and a 3.3-fold greater likelihood of prolonged P2B (> 12 hours) during lockdown as compared to 2019.

The delayed P2B time may share similar causes as the decrease in the incidence of STEMI. There was no change in the PPCI pathway or STEMI admission policy during the lockdown period. Admitted patients were tested for SARSCoV-2 according to the Ministry of Health guidelines but their admission and treatment were not delayed. However, the increased D2B time may be partly explained by the need of the medical staff to put on personal protective equipment, as well as emergency department (ED) increased workload (for those who arrived through ED). While these differences did not translate to a significant difference in in-hospital mortality (though a nominal increase was observed), both admission and peak troponin levels were higher, and more patients developed hemodynamic instability throughout their hospitalization in the 2020 patient group. The length of hospital stay was longer, and fewer patients were discharged early following PPCI despite a general tendency to shorten the length of hospital stay during the pandemic. These may serve as surrogate markers of severity, with an expected worse long-term prognosis.

Several limitations of this study should be acknowledged. First, we compared the contemporary cohort data to a retrospective one, with its inherent limitation. Second, our short

**Table 1. Patients' clinical characteristics and in hospital outcomes.**

|  | 2019 (n = 136) | 2020 (n = 107) | P value |
|---|---|---|---|
| Age, year Median (IQR) | 61 (51,68) | 63 (52,70 | 0.33 |
| Women, % | 18.4 | 15.9 | 0.61 |
| Ischemic Heart Disease, % | 33.8 | 27.1 | 0.26 |
| Diabetes Mellitus, % | 29.4 | 34.6 | 0.48 |
| Hypertension, % | 47.8 | 52.3 | 0.48 |
| Dyslipidemia, % | 53.7 | 58.9 | 0.42 |
| Smoker, % | 61 | 63.6 | 0.88 |
| Atrial Fibrillation, % | 6.6 | 7.5 | 0.79 |
| Family History of Ischemic Heart Disease, % | 17.8 | 24 | 0.24 |
| Infarct Related Artery (IRA) |  |  | 0.53 |
| LMCA | 2.2 | 2.8 |  |
| LAD | 47.4 | 36.8 |  |
| LCX/Ramus intermedius | 11.1 | 12.1 |  |
| RCA | 37.8 | 45.3 |  |
| SVG grafts | 1.5 | 1.9 |  |
| Multivessel Disease | 56.6 | 55.1 | 0.82 |
| Pre PCI TIMI flow in IRA, % |  |  | 0.26 |
| TIMI 0 | 54.1 | 50 |  |
| TIMI 1 | 8.9 | 5.7 |  |
| TIMI 2 | 12.6 | 21.7 |  |
| TIMI 3 | 24.4 | 22.6 |  |
| P2B, hours median (IQR) | 3 (2,5.75) | 4 (3,8.5) | 0.01 |
| D2B, Min median (IQR) | 49 (31,75) | 56 (30, 89) | 0.22 |
| D2B >90 min | 11.9 | 24 | 0.01 |
| P2B >12 hours | 7.6 | 19 | 0.01 |
| Inability to achieve TIMI 3 flow post PCI, % | 5.9 | 8.5 | 0.44 |
| Admission Troponin, ng/L median (IQR) | 54 (20,623) | 150 (43, 608) | 0.03 |
| Peak Troponin, ng/L median (IQR) | 2,648 (1,033, 6,300) | 4,365 (2,000, 10,000) | 0.01 |
| CCU length of stay, Days median (IQR) | 3 (3,4) | 4 (3,5) | 0.24 |
| Hospital length of stay, Days median (IQR) | 4 (3,6) | 5 (4,6) | 0.03 |
| Early Discharge, % | 32.4 | 12.4 | <0.001 |
| Mechanical Ventilation, % | 13.2 | 12.1 | 0.81 |
| Hemodynamic instability, % | 14.7 | 25.2 | 0.04 |
| VF/ Cardiac arrest, % | 12.5 | 14 | 0.73 |
| LVEF (, % median (IQR) | 42 (35, 55) | 43 (35,50) | 0.66 |
| Reduced LVEF,% | 46.6% | 46.6% | 0.99 |
| Mortality, % | 5.2 | 8.4 | 0.32 |

follow-up and the relatively small sample size do not allow us to conclude on hard endpoints such as mortality. However, it should be noted that the association between prolonged reperfusion times and outcomes is well established [11–13].

## Conclusion

The lockdown imposed in March -April 2020 during the SARSCoV-2 pandemic was associated with a fewer STEMI admission and a significant lag in the time intervals to reperfusion, higher troponin levels and longer hospital stays of STEMI patients. As reperfusion times

**Table 2. Adjusted\* odds ratio for delayed reperfusion times.**

| | Lockdown period 2020 | |
|---|---|---|
| | OR (95% CI) | P value |
| D2B>90 min | 2.4 (1.2–4.9) | 0.01 |
| P2B>12 hours | 3.3 (1.3–8.1) | <0.01 |

D2B: Door-to-Balloon, P2B: Pain-to-Balloon

Reference group: 2019 admissions; Adjusted to age, gender, ischemic heart disease, hypertension, Smoker, diabetes mellitus, and dyslipidemia. D2B: Door-to-Balloon, P2B: Pain-to-Balloon

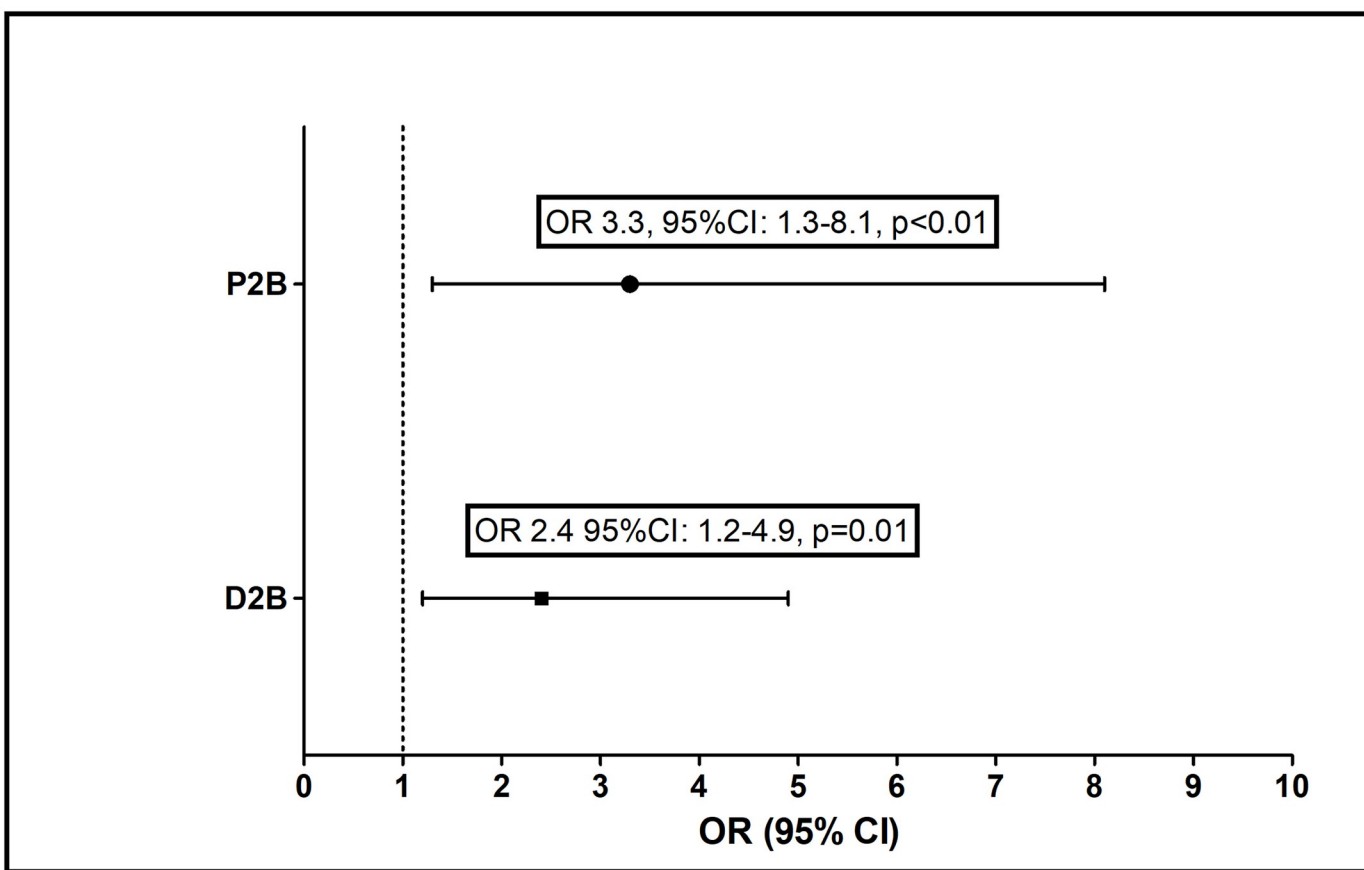

**Fig 1. Adjusted\* odds ratio for delayed reperfusion times.** * Reference group: 2019 admissions; Adjusted to age, gender, ischemic heart disease, hypertension, Smoker, diabetes mellitus, and dyslipidemia. D2B: Door-to-Balloon, P2B: Pain-to-Balloon.

directly affect the clinical outcomes of STEMI patients, measures to improve these metrics should be implemented prior to and during any future lockdown.

## Supporting information

**S1 Table. Weekly confirmed new SARS-CoV-2 cases in Israel during lockdown.**
(DOCX)

## Author Contributions

**Conceptualization:** Ofer Kobo, Roi Efraim, Sa'ar Minha.

**Data curation:** Majdi Saada, Ala Abu Dogosh.

**Methodology:** Ofer Kobo, Roi Efraim.

**Writing – original draft:** Ofer Kobo, Roi Efraim.

**Writing – review & editing:** Ofer Kobo, Majdi Saada, Natalia Kofman, Ala Abu Dogosh, Yigal Abramowitz, Doron Aronson, Sa'ar Minha, Ariel Roguin, Simcha R. Meisel.

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
