## [Decision Letter · Decision Letter 0]

18 Sep 2020

PONE-D-20-24693

The impact of lockdown enforcement during the SARSCoV-2 pandemic on the number and timing of presentation of patients with ST-elevation myocardial infarction

PLOS ONE

Dear Dr. kobo,

Thank you for submitting your manuscript to PLOS ONE. After careful consideration, we feel that it has merit but does not fully meet PLOS ONE’s publication criteria as it currently stands. Therefore, we invite you to submit a revised version of the manuscript that addresses the points raised during the review process.

We look forward to receiving your revised manuscript.

Kind regards,

Corstiaan den Uil

Academic Editor

PLOS ONE

Journal Requirements:

2.In the methods section please provide further information regarding 1) how information on variables reported  was obtained and 2) inclusion and exclusion criteria.

Furthermore, this is a retrospective study with no control group. As such, we do not feel that any conclusions on the intervention effects can be supported; as such, we ask that you revise the text (especially, but no limited to, the aims and Conclusions) to avoid unsupported statements.

3.We note that you have indicated that data from this study are available upon request. PLOS only allows data to be available upon request if there are legal or ethical restrictions on sharing data publicly. For information on unacceptable data access restrictions, please see http://journals.plos.org/plosone/s/data-availability#loc-unacceptable-data-access-restrictions.

Reviewers' comments:

Reviewer's Responses to Questions

**Comments to the Author**

1. Is the manuscript technically sound, and do the data support the conclusions?

Reviewer #1: Partly

Reviewer #2: Yes

2. Has the statistical analysis been performed appropriately and rigorously? 

Reviewer #1: I Don't Know

Reviewer #2: Yes

3. Have the authors made all data underlying the findings in their manuscript fully available?

Reviewer #1: Yes

Reviewer #2: Yes

4. Is the manuscript presented in an intelligible fashion and written in standard English?

Reviewer #1: Yes

Reviewer #2: Yes

5. Review Comments to the Author

Reviewer #1: The authors present an observational paper exploring the impact in Israel of the enforced lockdown during the SARSCoV-2 pandemic on the characteristics of presentation of STEMI patients. STEMI patients admitted during the same time period in 2019 served as the control group. A total of 342 patients were included from 4 cardiac centres. The major findings reported were i) a 22% decrease in the number of STEMI admissions, ii) 2.4-fold greater likelihood of prolonged door-to-balloon time (> 90 min), and iii) 3.3-fold greater likelihood of prolonged pain-to-balloon time (> 12 hours) post lockdown as compared to 2019. Consistent with delayed time to reperfusion; peak troponin and the incidence of hemodynamic instability were higher in the 2020 cohort. There was no difference in the baseline characteristics of the patients.

While the paper was of interest, there are a number of issues that should be addressed.

A] The title of the paper is “The impact of lockdown enforcement during the SARSCoV-2 pandemic on the number and timing of presentation of patients with ST-elevation myocardial infarction”. However, for some of the analyses the authors include in the 2020 cohort patients admitted prior to when lockdown measures were introduced in Israel (20th March). What is the purpose of this? Subsequently they then compare a pre-lockdown 2020 as well as post-lockdown cohort to the 2019 control group for outcomes such as reperfusion times. This is confusing and does not appear to add to the paper. Was there a particular reason for this approach? If so, please make this clearer in the manuscript.

B] To better interpret the data and impact of lockdown on study measures, it would be helpful to include information on the weekly incidence rate of COVID-19 cases in Israel during the 2020 study period.

C] The authors stated the incident rate for STEMI admissions post-lockdown fell by 22% (RIR 0.78 with a p value of 0.05) compared to 2019. What statistical method was used for comparison?

D] What defined a STEMI admission? Was is someone who underwent PPCI? If so, how do the authors know a fall in STEMI admissions was not driven by a change in threshold to refer / admit to the cath-lab? During the post-lockdown period was there a change to STEMI / PPCI pathways?

E] The authors state that only one SARS-CoV-2 positive patient was included in the study. How many of the 2020 cohort were tested for SARS-CoV-2?

F] With respect to pain to balloon time, door to balloon time and peak troponin, what percentage of patients was data available?

G] Was there a change in the PPCI pathway (e.g. admission / screening through the Emergency Department prior to the cath lab rather than direct to the cath lab) that may account for the increase in door to balloon times observed?

H] Please define the criteria for “haemodynamic instability”.

I] It should be made clearer on the Figure that the odds ratio is

F] In the Introduction the authors state “While early reports described decreased admissions for acute cardiovascular event during the SarsCoV-2 pandemic, the impact of governmental restriction measures on patient outcomes has not been evaluated.” This is not the case. Multiple papers from Europe (including Italy, Spain and the UK) have already looked at the impact of governmental restrictions on patient outcomes. Many of these papers are already referenced in the manuscript. This sentence should be amended (along with a similar sentence in the Abstract).

G] Please replace all NS for p-values with the actual number.

H] The first paragraph of the Discussion is poorly written with too much conjecture. Many of these arguments have already been made in similar published manuscripts and the whole paragraph spends too much copy not adding any original perspective (and indeed references) to the debate as to why STEMI admissions fell during the early phase of COVID-19.

I] The conclusion states: “this study reveals the direct influence of the lockdown restrictions on public health issues and brings to attention the hazard of delayed reperfusion of STEMI patients.” There was no difference in LVEF or mortality. Accordingly, I don’t think the conclusion this study brings to attention the hazard of delayed reperfusion in STEMI is justified nor would it seem to summarise the most important findings of the paper.

J] The paper is reasonably well written but suffers from a number of important grammatical errors that should be amended. For example, the final sentence of the Conclusion “As this it well associated with poor outcome, and some countries and region experience second Covid 19 wave, measures to improve this metric should be implemented prior to any future lockdown” is not well written and could be improved.

In addition to above, the major shortcoming of this paper is that many of the findings have already been published. This is especially the case with respect to the change in incidence of STEMI and baseline characteristics. Published data is already available from much larger datasets and the present manuscript would not seem to add much in this regard. However, there is substantially less data about the impact of COVID-19 (or lockdown) on reperfusion times (currently published studies: Secco GG et al. Decrease and Delay in Hospitalization for Acute Coronary Syndromes during the 2020 SARS-CoV-2 Pandemic. Can J Cardiol. 2020. Tam C-CF et al. Impact of Coronavirus Disease 2019 (COVID-19) Outbreak on ST-Segment-Elevation Myocardial Infarction Care in Hong Kong, China. Vol. 13, Circulation. Cardiovascular quality and outcomes. 2020. Wilson, S. J. et al. Effect of the COVID-19 Pandemic on ST-Segment–Elevation Myocardial Infarction Presentations and In-Hospital Outcomes. Circulation: Cardiovascular Interventions, 13(7). 2020). This would seem to be the major value of this study in terms of adding to the literature. A short report centred on the effect of lockdown measures in Israel on reperfusion times, troponin and other in-hospital outcomes (with 2 clearly defined cohorts: post-lockdown versus calendar-matched 2019 cohort) would seem far more attractive.

Reviewer #2: The study describes the impact of the lockdown on the admission of STEMI stating that it is the first time such data are reported. In fact, many reports have already reported similar results worldwide. Even concerning the impact of the lockdown, it has been already demonstrated for french patients (Lantelme et al ; Arch Cardiovasc Dis Jun-Jul 2020;113(6-7):443-447. This paper does not bring new findings.

The authors stated that the study was performed in 4 high volume University hospitals with PCI facilities. Often such high volume centres were overflowed with severe COVID-19 patients who require most of intensive care facilities. Could it be that some patients with an acute coronary syndrome were directed to other hospitals during this period ? How can the authors rule out this hypothesis ?

Overall, the number of STEMI was rather stable with a sort of redistribution of patients: more STEMI were referred before the lockdown as compared to 2019 and less after. How do the authors explained this increase pre-lockdown referral ?

6. PLOS authors have the option to publish the peer review history of their article (what does this mean?). If published, this will include your full peer review and any attached files.

Reviewer #1: No

Reviewer #2: No

---

## [Author Response · Author response to Decision Letter 0]

7 Oct 2020

Reviewer 1:

A] The title of the paper is “The impact of lockdown enforcement during the SARSCoV-2 pandemic on the number and timing of presentation of patients with ST-elevation myocardial infarction”. However, for some of the analyses the authors include in the 2020 cohort patients admitted prior to when lockdown measures were introduced in Israel (20th March). What is the purpose of this? Subsequently they then compare a pre-lockdown 2020 as well as post-lockdown cohort to the 2019 control group for outcomes such as reperfusion times. This is confusing and does not appear to add to the paper. Was there a particular reason for this approach? If so, please make this clearer in the manuscript.

Response:

According to this and other comments we revised the manuscript to include the lockdown period only (compared to same period in 2019). 

B] To better interpret the data and impact of lockdown on study measures, it would be helpful to include information on the weekly incidence rate of COVID-19 cases in Israel during the 2020 study period.

Response:

Thank you for your suggestion. Table S1 now include the weekly confirmed new SARS-CoV-2 cases in Israel during Lockdown

C] The authors stated the incident rate for STEMI admissions post-lockdown fell by 22% (RIR 0.78 with a p value of 0.05) compared to 2019. What statistical method was used for comparison?

Response:

Thank you, as we omitted the pre-lockdown period, RIR calculations were also omitted from the revised manuscript. 

D] What defined a STEMI admission? Was is someone who underwent PPCI? If so, how do the authors know a fall in STEMI admissions was not driven by a change in threshold to refer / admit to the cath-lab? During the post-lockdown period was there a change to STEMI / PPCI pathways?

Response:

In the revised methods section, we defined STEMI admission as requested. In the discussion methods we specified that there was no chance in the PPCI pathway and STEMI admission policy during lockdown.

E] The authors state that only one SARS-CoV-2 positive patient was included in the study. How many of the 2020 cohort were tested for SARS-CoV-2?

Response:

Unfortunately we do not have this data, we did not collect this data as this was not a pre-defined outcome. 

F] With respect to pain to balloon time, door to balloon time and peak troponin, what percentage of patients was data available?

Response:

This data was available to 100% of the patients. 

G] Was there a change in the PPCI pathway (e.g. admission / screening through the Emergency Department prior to the cath lab rather than direct to the cath lab) that may account for the increase in door to balloon times observed?

Response:

Please see respond to “D” – there was no change in the pathway or admission policies.

H] Please define the criteria for “haemodynamic instability”.

Response:

Thank you, haemodynamic instability is now clearly defined in the revised methods section.

I] It should be made clearer on the Figure that the odds ratio is

Response:

OR labels were added to the figure as suggested, thank you

F] In the Introduction the authors state “While early reports described decreased admissions for acute cardiovascular event during the SarsCoV-2 pandemic, the impact of governmental restriction measures on patient outcomes has not been evaluated.” This is not the case. Multiple papers from Europe (including Italy, Spain and the UK) have already looked at the impact of governmental restrictions on patient outcomes. Many of these papers are already referenced in the manuscript. This sentence should be amended (along with a similar sentence in the Abstract).

Response:

Thank you for your justified comment, the sentence was amended.

G] Please replace all NS for p-values with the actual number.

Response:

Revised as requested, see table 1.

H] The first paragraph of the Discussion is poorly written with too much conjecture. Many of these arguments have already been made in similar published manuscripts and the whole paragraph spends too much copy not adding any original perspective (and indeed references) to the debate as to why STEMI admissions fell during the early phase of COVID-19.

Response:

Thank you for your comment, we shortened and edited the paragraph as suggested. 

I] The conclusion states: “this study reveals the direct influence of the lockdown restrictions on public health issues and brings to attention the hazard of delayed reperfusion of STEMI patients.” There was no difference in LVEF or mortality. Accordingly, I don’t think the conclusion this study brings to attention the hazard of delayed reperfusion in STEMI is justified nor would it seem to summarise the most important findings of the paper.

Response:

Conclusion section was revised to better reflect the main findings of the study.

J] The paper is reasonably well written but suffers from a number of important grammatical errors that should be amended. For example, the final sentence of the Conclusion “As this it well associated with poor outcome, and some countries and region experience second Covid 19 wave, measures to improve this metric should be implemented prior to any future lockdown” is not well written and could be improved.

Response:

Thank you for your comment. The revised manuscript underwent language editing. 

In addition to above, the major shortcoming of this paper is that many of the findings have already been published. This is especially the case with respect to the change in incidence of STEMI and baseline characteristics. Published data is already available from much larger datasets and the present manuscript would not seem to add much in this regard. However, there is substantially less data about the impact of COVID-19 (or lockdown) on reperfusion times (currently published studies: Secco GG et al. Decrease and Delay in Hospitalization for Acute Coronary Syndromes during the 2020 SARS-CoV-2 Pandemic. Can J Cardiol. 2020. Tam C-CF et al. Impact of Coronavirus Disease 2019 (COVID-19) Outbreak on ST-Segment-Elevation Myocardial Infarction Care in Hong Kong, China. Vol. 13, Circulation. Cardiovascular quality and outcomes. 2020. Wilson, S. J. et al. Effect of the COVID-19 Pandemic on ST-Segment–Elevation Myocardial Infarction Presentations and In-Hospital Outcomes. Circulation: Cardiovascular Interventions, 13(7). 2020). This would seem to be the major value of this study in terms of adding to the literature. A short report centred on the effect of lockdown measures in Israel on reperfusion times, troponin and other in-hospital outcomes (with 2 clearly defined cohorts: post-lockdown versus calendar-matched 2019 cohort) would seem far more attractive.

Response:

Thank you for this thoughtful comment. After discussion we felt that we should revised the manuscript as suggested to focus on the reperfusion times and early clinical outcome during lockdown period, as suggested.

Reviewer #2: The study describes the impact of the lockdown on the admission of STEMI stating that it is the first time such data are reported. In fact, many reports have already reported similar results worldwide. Even concerning the impact of the lockdown, it has been already demonstrated for french patients (Lantelme et al ; Arch Cardiovasc Dis Jun-Jul 2020;113(6-7):443-447. This paper does not bring new findings.

The authors stated that the study was performed in 4 high volume University hospitals with PCI facilities. Often such high volume centres were overflowed with severe COVID-19 patients who require most of intensive care facilities. Could it be that some patients with an acute coronary syndrome were directed to other hospitals during this period ? How can the authors rule out this hypothesis ?

Response:

Thank you for your comment. During March-April lockdown intensive care units in our centers were not overflowed and ACS patients were not diverted or directed to other hospitals.

Overall, the number of STEMI was rather stable with a sort of redistribution of patients: more STEMI were referred before the lockdown as compared to 2019 and less after. How do the authors explained this increase pre-lockdown referral ?

Response:

As we revised the manuscript according to Reviewer #1 suggestions , the pre-lockdown period is not included in the revised manuscript

---

## [Editor Report · Decision Letter 1]

9 Oct 2020

The impact of lockdown enforcement during the SARSCoV -2 pandemic on the timing of presentation and early outcomes of patients with ST-elevation myocardial infarction

PONE-D-20-24693R1

Dear Dr. kobo,

We’re pleased to inform you that your manuscript has been judged scientifically suitable for publication and will be formally accepted for publication once it meets all outstanding technical requirements.

Kind regards,

Corstiaan den Uil

Academic Editor

PLOS ONE
---

## [Editor Report · Acceptance letter]

16 Oct 2020

PONE-D-20-24693R1 

The impact of lockdown enforcement during the SARSCoV-2 pandemic on the timing of presentation and early outcomes of patients with ST-elevation myocardial infarction 

Dear Dr. Kobo:

I'm pleased to inform you that your manuscript has been deemed suitable for publication in PLOS ONE. Congratulations! Your manuscript is now with our production department. 

Kind regards, 

on behalf of

Dr. Corstiaan den Uil 

Academic Editor

PLOS ONE